# Liquidity effects on oil volatility forecasting: From fintech perspective

**Shusheng Ding[1], Tianxiang Cui[2]\*, Yongmin Zhang[1], Jiawei Li[2]**

1 School of Business, Ningbo University, Ningbo, Zhejiang, China, 2 School of Computer Science, University of Nottingham Ningbo China, Ningbo, Zhejiang, China

\* tianxiang.cui@nottingham.edu.cn

**Data Availability Statement:** The data used in this study are third party data owned by Thomson Reuters. The authors accessed the data using an account paid for by their university. Others may access this data at the Datastream database of Thomson Reuters [https://www.thomsonone.com/

## Abstract

Fin-tech is an emerging field, inspiring revolutionary innovations in the financial field. It may initiate the evolutionary episode of the financial research, where volatility forecasting is a crucial topic in finance. For forecasting volatility, GARCH model is a prevailing model, however, further improvement of the GARCH model is still challenging. In this paper, we demonstrate how Fintech can play a part in volatility forecasting by employing a metaheuristic procedure called Genetic Programming. On the basis, we are able to develop a new volatility forecasting model, which can beat GARCH family models (including GARCH, IGARCH and TGARCH models) in a significant way. Since genetic programming is an evolutionary algorithm based on the principles of natural selection, this innovative work will be a breakthrough point in the financial area. The innovation of this paper demonstrates how GP technology can be applied in the financial field, attempting to explore the volatility forecasting area from the combination of new technology and finance, known as fintech. More importantly, when the formula of volatility forecasting is unknown as we introduce a new factor, namely, the liquidity factor, we unveil that how GP method can be helpful in determining the specific volatility forecasting model format. We thereby exhibit the liquidity effects on volatility forecasting filed from the fintech perspective.

## 1 Introduction

### 1.1 Motivations and aims

Fin-tech is an emerging field, which drives revolutionary innovations in the financial spectrum recently. It integrates advanced technology into financial area, provoking the evolutionary episode of the financial industry and research (Buchak et al. 2018 [1]; Chen et al. 2019 [2]). Fintech has been playing particularly important roles in two areas of financial innovations: derivative trading and high frequency trading, which demand sophistical mathematical models, computing-intensive algorithms and big data handling techniques in order to capture trading signal in microseconds. In the option markets, volatility is the dominant driving factor for option price moves. In high frequency trading, volatility forecasting is essential to manage portfolio risk. Therefore, volatility forecasting would be a key issue in the financial field and

DirectoryServices/2006-04-01/Web.Public/Login.
aspx?brandname=datastream]. The data set was
extracted from the time series section of the
Datastream database for the US crude oil futures
market daily data, ranging from January 1, 2001 to
December 31 2019. The authors confirm that they
did not have special access privileges that others
would not have.

**Funding:** The author(s) received no specific
funding for this work.

**Competing interests:** The authors have declared
that no competing interests exist.

volatility forecasting models have extensive applications, such as market timing, portfolio selection, as well as financial risk management (Lien and Wilson 2001 [3]).

GARCH models are prevalent in forecasting volatilities of financial markets (see Andersen et al. 2005 [4]; Tian and Hamori [5]) and oil market volatility has been intensively examined and forecasted by adopting GARCH models (see Wei et al. 2010 [6]; Efimova and Serletis 2014 [7]; Wang and Liu 2016 [8]; Xing and Wang 2019 [9]). However, existing GARCH type models are unable to capture either market micro structures, such as liquidity information or the high frequency trading signals because the volatility estimation precision from GARCH models depends on the accuracy of mean estimation models, which are in low frequency basis. Incorporating liquidity into GARCH models is necessary in order to estimate volatility more accurately, especially from high frequency trading.

In fact, the relationship between liquidity and volatility has been scrutinized in the existing literature. Fleming and Remolona (1999) [10] have investigated the relation between liquidity, volatility and public information in the US Treasury market. They declare that volatility and liquidity respond to the public information simultaneously. Feng, Hung and Wang (2014)'s [11] empirical results also support the liquidity impact on the volatility. More recently, Collin-Dufresne and Fos (2016) [12] explore the relations between liquidity and noise trading volatility. They affirm the significant impact of liquidity on noise trading volatility.

Volatility forecasting by using Artificial Intelligence (AI) and Genetic Programming (GP) can been witnessed in the literatures (see Yin et al. 2016; [13]; Ding et al. 2019 [14]; Weng et al. 2021 [15]; Mademlis and Dritsakis 2021 [16]). The goal of the paper is to adopt AI technologies to generate the best model which can comprise trading liquidity effect into volatility forecasting, and can be integrated into the existing fintech systems such as high frequency trading platforms and derivative trading platforms in the hedge fund industry.

## 1.2 Research contributions

The first contribution is that we adopt GP to identify the model format for forecasting oil volatility with liquidity. It is arguable that liquidity has a considerable effect on market volatility but how to integrate liquidity factor into the volatility forecasting model has no unanimous solution. Our GP system can identify the specific format of volatility forecasting with liquidity by two steps: (i) model formation search and (ii) model accuracy evolution.

In the first stage, we identify the potential form of volatility forecasting model. The GP is used to search the potential forms of the volatility forecasting model with integration of liquidity variable. GP holds the elegant characteristics that one can build the relevant performance criterion directly into the search procedure.

In the second stage, we apply Darwin's theory of evolution to model accuracy evolution. To be specific, we adopt crossover and mutation operations during our volatility forecasting model selection and development. The main advantage of crossover and mutation algorithms is that they can help models to evolve according to the historical data in order to minimize forecasting errors. As a result, the generated forecasting model can provide far better forecasting results after massive generations of model evolution.

Based on this process, our GP system can identify the most relevant terms that have effects on predicting oil volatility. The bid ask spread term has been included in the volatility forecasting model by GP system, which captures the liquidity effect as it provides the trading information of the financial market (Deuskar et al. 2011 [17]). Since the volatility forecasting model format with liquidity is unknown, our contribution relies on the specific format identification of volatility forecasting model with liquidity, which makes accurate forecasting and liquidity sensitivity analysis applicable for the oil market.

Furthermore, our work is related heavily to existing literature where GP has been adopted in various financial areas. For example, Pimenta et al. (2018) [18] use GP to develop automated investing method in the stock market. More recently, Michell and Kristjanpoller (2020) [19] employ GP to develop trading rules in the US stock market. Ding et al. (2020) [20] apply GP to forecast future stock returns in different stock markets. Our paper extends the GP application to the oil market volatility forecasting, which is a crucial commodity market.

Forecasting oil price volatility is not only useful to commodity traders for speculating and hedging (Ma et al. 2019 [21]), but also important to economists for forecasting macroeconomic variables, such as inflation and industrial productions (Yin 2016 [22]; Chen et al. 2019 [23]). After the model was generated by our algorithm, we compare our model with GARCH models regarding the forecasting ability. The empirical results show that the in-sample performance of our model is more accurate than traditional GARCH models for both full-sample and most subsample predictions. Additionally, our model's out-of-sample volatility prediction is also more precise compared with traditional GARCH models for both full-sample and subsample tests. The accurate volatility prediction of our GARCH can be envisioned as a considerable improvement of the existing GARCH models and our results demonstrate that multivariable GARCH model can still deliver accurate volatility predictions.

Finally, our research also produces significant empirical contributions. We shed new insights on the concept of "Fin-tech". The term "artificial intelligence" shall be a pivotal ingredient in the fintech concept, which refers when a machine mimics cognitive functions that humans associate with other human minds, such as "learning" and "problem solving" (Russell and Norvig 2009 [24]). However, the prevailing recognition of fintech focuses on the application of technology into the financial service industry, such as online banking and mobile payment (Mackenzie 2015 [25]). This paper gives a further annotation, suggesting that fintech also involves how AI, such as genetic programming, can be integrated with financial modelling and this paper itself is a magnificent demonstration of finance with AI, which is a radical innovation in finance. This integration can provide deep insights of financial data analysis since all models are generated and evaluated purely based on financial data. Furthermore, our algorithm is also capable of dealing with big data as well as other forecasting problems, such as return forecasting. Consequently, this paper can serve as an impetus of developing other fintechs in hedge fund management including derivatives trading algorithms and high-frequency trading.

The remainder of the paper is structured as follows. In section 2, we describe the data and variables used in the paper. In section 3, we derive our GARCH model with the GP method. Section 4 presents the empirical results and the model performance in data fitting and volatility forecasting. Section 5 delivers the conclusions and implications of the paper.

## 2 Data and variable estimation

### 2.1 The data

The GARCH(1,1) model has been widely used in the financial literatures for volatility forecasting (see Bollerslev and Wright 2001 [26]; Hansen and Lunde 2005 [27]), especially in the oil market (see Klein and Walther 2016 [28]; Lux et al. 2016 [29]). As a consequence, we also use our model to forecast the oil return volatility compared with the traditional GARCH model and GARCH family models (see Fig 1 for oil daily returns). We will use the univariate GARCH-class models, including GARCH, IGARCH and the GARCH model with asymmetric effects, namely, GJR-GARCH, as benchmark models for oil volatility predictions in our paper. The liquidity effect would be measured by the bid-ask spread (BAS), which has been demonstrated to be positively correlated with price volatility in financial markets (see Bollerslev and

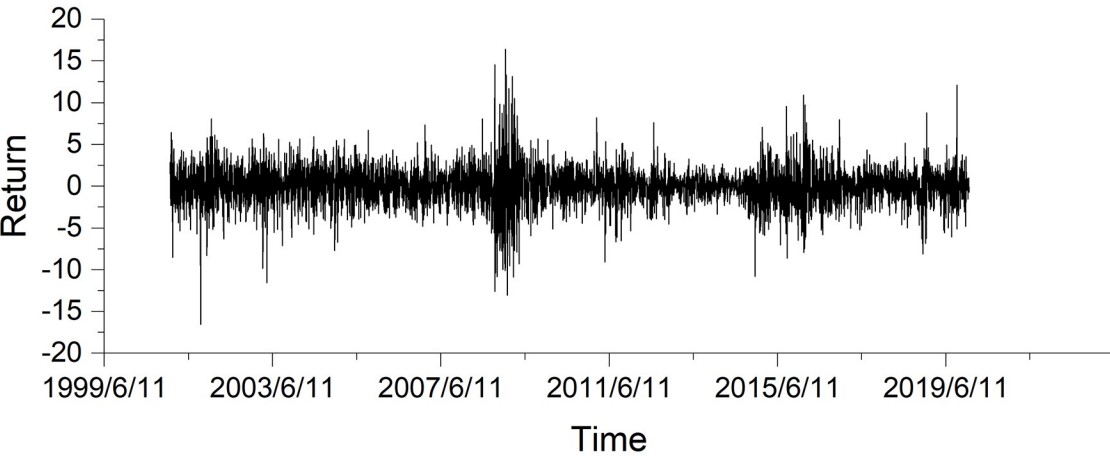

**Fig 1. Daily return plotting.**

Melvin 1994 [30]; Haugom et al. 2014 [31]). We have not included the weekend effect in this paper, which is popular in the stock markets. Studies on the weekend and daily effect in oil futures market are scant since it has been documented that the weekend effect may not be obvious in oil futures market (Yu and Shin 2011 [32]). It might be that oil futures contract has time-to-maturity, which may have a stronger effect than the weekend effect (Geman and Kharoubi, 2008 [33]).

We obtain the WTI oil futures trading data from Thomson Datastream with daily frequency, from January 1, 2001 to December 31 2019, yielding a total of 4,752 trading days with 4,752 observations. Most literatures have adopt such database and the trading data is continuous, which would be favored for daily volatility forecasting. We use the period from January 1, 2001 to December 31, 2010 as the in-sample period and the period from January 1, 2011 to December 31, 2019 as the out-of-sample period. By "out-of-sample", we imply two things: Firstly, we use the previous year's data to estimate coefficients of the model, which forecasts this year's volatility, namely, we adopt one-year rolling window for the test.

For in-sample fitting, the estimation year and prediction year are the same. More importantly, we only use the period from January 1, 2001 to December 31, 2010 as the sample period for genetic programming to generate our model. We derive the theoretical model from genetic programming and we estimate all the model coefficients by regression. Then, we apply the generated volatility forecasting model to the period of January 1, 2011 to December 31, 2019 to testify the "out-of-sample" model performance. For the out-of-sample forecasting, we use previous year's data to forecast the volatility of this year. For example, we use the data of 2013 to estimate the model parameters and then use those parameters to forecast volatility of year 2014. Similarly, we use the data of 2014 to estimate the model parameters and then use those parameters to forecast volatility of year 2015. We thereby roll over the sample by one year.

Furthermore, we use two whole periods as the full-sample volatility forecasting test and we use one year within the whole period as the subsample forecasting tests. In addition, for both in-sample and out-of-sample tests, we use 1-day ahead prediction during the data period and the statistical test for error differences will be also employed.

## 2.2 Variable estimation

Because the specific computational technique we have adopted, it may not allow us to use two identical time series data even with different time lags. As a result, we adopt two different time

series for modeling the variance. The first sequence is the daily simplified realized variance series. Since we adopt a relatively low frequency data, we use the low frequency daily simplified realized variance rolling average in one month to approximate the daily simplified realized variance(see Schwert 1989 [34]), which is denoted as $SRV_t$ and can be defined as:

$$SRV_t = \frac{1}{T}\sum_{i=1}^{T}(r_{t-i})^2 * 10,000, \tag{1}$$

since the mean of daily realized return is near 0 with the realized return $r_t$ is defined as:

$$r_t = \ln\left(\frac{P_t}{P_{t-1}}\right) * 100$$

(Andersen and Bollerslev 1998 [35]), where $P_t$ is the last day's settlement price and we enlarged the SRV measurement by 10,000 times and the return by 100 times to avoid extreme small values.

The second sequence is the realized daily variance, which is denoted as $RV_t$ and we use the standard deviation of the oil futures daily return (Christensen and Prabhala 1998 [36]), which can be defined as:

$$RV_t = \frac{1}{T-1}\sum_{i=1}^{T}(r_{t-i} - \bar{r})^2 * 10,000, \tag{2}$$

where

$$\bar{r} = \frac{1}{T-1}\sum_{i=1}^{T} r_{t-i}.$$

We enlarged the measurement by 10,000 times to avoid extreme small values.

For both realized daily variance and simplified realized daily variance, we use the one-month rolling average approach and we use the updated real market data for each day prediction without iteration. We then define the simplified realized variance as the target series to be forecasted and we use other three series to fit the data, namely, Roll spread, $\varepsilon_t$ and realized variance where $\varepsilon_t$ is the residual taken from GARCH(1,1) model. As the BAS is positively related with the volatility, we use the Roll measure to approximate the daily BAS measurement. The Roll spread is an effective spread estimator for the bid-ask spread calculation and suits the low frequency data (Marshall et al. 2012 [37]; Haugom et al. 2014 [31]) and the Roll measure thereby could be severed as the market liquidity proxy. Specifically, Roll (1984) [38] assumes that the stocks have fundamental values, denoted as $V_t$ at time $t$ and $V_t$ follows:

$$V_t = V_{t-1} + \eta_t \tag{3}$$

$\eta_t$ is the residual term for this fundamental value random process, which differentiates from the term $\varepsilon_t$, which is the residual taken from GARCH volatility model, and $\eta_t$ is the mean-zero, serially uncorrelated public information shock on day $t$.

Next, he denotes $S_t$ as the last observed trade price on day $t$ and presumes that $S_t$ follows:

$$S_t = V_t + \frac{1}{2}EQ_t,$$

where $E$ is the effective spread and $Q_t$ is a buy/sell indicator for the last trade that equals +1 for a buy and -1 for a sell. He further assumes that $Q_t$ equally likely to be +1 or -1 and $Q_t$ is also serially uncorrelated, and independent of $\eta_t$.

**Table 1. Statistical summary of variables used for the 18 year data.**

| Variable | Obs | Mean | Std. Dev. | Min | Max |
|---|---|---|---|---|---|
| BAS | 4,752 | 0.6518 | 0.8023 | 0 | 8.111 |
| Return | 4,752 | 0.01731 | 2.312 | -16.54 | 16.40 |
| Simplified realized variance | 4,752 | 535.11 | 632.18 | 29.85 | 6037.48 |
| Realized variance | 4,752 | 21083.9 | 9649.8 | 5104.7 | 79452.2 |
| Price | 4,752 | 63.5942 | 25.8163 | 17.45 | 145.29 |

Then he takes the first difference of equation and plugs in the result from equation, which yields

$$\Delta S_t = \frac{1}{2} E \Delta Q_t + \eta_t \qquad (4)$$

As a result, $\text{cov}(\Delta S_t, \Delta S_{t-1}) = -\frac{1}{4} E^2$ or equivalently,

$$spread = 2\sqrt{-\text{cov}(\Delta S_t, \Delta S_{t-1})}.$$

Because when the auto-covariance is positive, the formula is undefined. We therefore use a modified version of the Roll estimator (Goyenko et al. 2009 [39]):

$$spread = \begin{cases} 2\sqrt{-\text{cov}(\Delta S_t, \Delta S_{t-1})}, \text{cov}(\Delta S_t, \Delta S_{t-1}) \leq 0 \\ 0, \text{cov}(\Delta S_t, \Delta S_{t-1}) > 0 \end{cases}. \qquad (5)$$

In order to estimate the spread, we first take the difference of daily price, then we take the covariance of the price difference of t and t-1. Then, we take the square root of the negative covariance if it is greater than 0 and we take the value of 0 otherwise.

To sum up, the simplified realized variance, the realized variance, the BAS and the residual term are the four main variables concerned by this study and the residual term $\varepsilon_t$ contains the noise information that has not been captured by the model. Table 1 presents the statistical summary of the main variables used in the paper.

## 3 Model development under genetic programming system

### 3.1 Preliminaries

In this section, we will develop our GARCH model based on the estimated variables in section 2. For the specific model development, we will adopt the Genetic Programming (GP) method from computer science, which is proposed by Koza (1994) [40]. GP is an evolutionary computation (EC) technique inspired by biological process Banzhaf et al. 1998 [41]; Hirsh et al. 2000 [42]; Poli et al. 2008 [43]). Since the form of volatility forecasting model with liquidity effects is uncertain, it would be beneficial to adopt GP method. One big advantage of adopting GP is that it can allow one to be agnostic about the general form of the model. In GP, a population of computer programs is evolved based on the principles of natural selection originated from Darwin's theory of evolution. After certain number of generations, GP can transform populations of programs into new and better programs. As stated in Poli et al. (2008) [43], GP has been very successful at evolving novel and unexpected ways of solving problems.

Our GP system proceeds in the following way. The system will create a number of functions as a population randomly. It firstly generates a random population of functions, and then the evaluations of every function will be performed by the system, where the forecasting accuracy

will be compared with the targeted function. We define the performance of the function as the fitness of the function.

Afterwards, genetic operations automatically choose one or two function(s) according the fitness evaluation. The genetic operations include two formats, namely, crossover and mutation. The crossover operation recombines two subitems from picked functions to produce new functions. The mutation operation will modify the subitems from picked functions before recombine them together to produce new functions. Then, based on crossover and mutation operations, the GP system will automatically reevaluate the newly-generated functions according to the function fitness. The probability of performing crossover and mutation operations will be pre-determined in the system. This main perquisites are the principles of evolution and the system will be discontinued after some pre-determined conditions are complied such as pre-determined threshold for function fitness. The system automatically chooses the best function as the solution according to the function fitness, which creates the new volatility forecasting model.

### 3.2 Genetic programming system

For our model development, we employ the following targeted functions as our forecasting task based on our GP system regarding the data sample period from January 1, 2001 to December 31, 2010:

$$f(L^2, \sigma^2, \varepsilon^2) = r^2 \tag{6}$$

where $L^2$, $\sigma^2$, $\varepsilon^2$ are squared BAS, realized variance and squared residuals respectively, and $r^2$ denotes the simplified realized variance, known as the squared return term, which is used to approximate the volatility. Our goal is to find the most relevant terms that have effects on predicting the simplified realized variance. For the robust purpose, we also adopt our model for realized variance forecasting, which has been defined in Section 2. The empirical forecasting results for both simplified realized variance and realized variance will be presented in Section 4.

Our GP system consists of the following parts:

- *Terminal Set*: $L^2$, $\sigma^2$, $\varepsilon^2$.

- *Function Set*: $+, -, \times$.

- *Fitness measure*: the error between the value of the individual function and the corresponding desired output (i.e. $r^2$).

- *GP parameters*: population = 10000, the maximum length of the program = 1000, probability of crossover operation = 0.8 and probability of mutation operation = 0.1.

- *Ending conditions*: when the measure of function fitness approximates 0 or the system runs up to 100 generations, the system will automatically discontinue. (For this work, the fitness measure will never reach 0, therefore the system will terminate after 100 generations.)

As the number of generation increases, the average error between the value obtained by forecasting functions and the target value may decrease. In this case, GP may produce some high-order terms in order to further reduce the error. However, these high-order terms may cause the overfitting problem. Therefore, we further eliminate the forecasting functions with more than 10 terms obtained by our GP system in order to prevent the overfitting. The parameters settings used above and the choice of 10 terms are based on empirical experiences and we

do not claim that these are the optimal choices. Our main purpose is to test the effectiveness of GP approach on volatility forecasting task.

### 3.3 Model development

With the settings stated above, we ran our GP system for 50 times in order to acquire the specific model with no $\varepsilon^2$ term. Since the $\varepsilon^2$ data series is in the program, the model solved by the program will contain the information incorporated in the term $\varepsilon^2$. Therefore, the model we devise with no $\varepsilon^2$ term can lead to the comprehensive decomposition of the $\varepsilon^2$ term for the model. After simplification, the best function with no $\varepsilon^2$ term we obtained is the following:

$$\sigma_t^2 = \alpha_0 + \alpha_1 \sigma_{t-1}^2 + \alpha_2 K_{t-1} \tag{7}$$

where $K_t = \sigma_t^2(1 - L_t^2) * (L_t^2 - \sigma_t^2 - L_t^4)$ and L presents the BAS. We name the model as LGARCH (1,1), which is a liquidity-adjusted GARCH model.

Furthermore, the GARCH (1,1) model has the following the form:

$$\sigma_t^2 = \alpha_0 + \alpha_1 \sigma_{t-1}^2 + \alpha_2 \varepsilon_{t-1}^2,$$
$$r_t = \phi + \varepsilon_t, \tag{8}$$

where $\sigma_t$ is the volatility of target time series and $\varepsilon_t$ is the residual term from the return prediction equation, which is: $\phi$ is the conditional mean, and $\varepsilon_t \sim N(0, \sigma_t^2)$.

Since the conditional variance may depend on the past squared residuals of the process, namely, $\varepsilon_t$, under the GARCH model, the $\varepsilon_t$ term plays a vital role in data fitting and conditional variance prediction. The general decomposition of $\varepsilon_t$ has been provided in the literatures, which is $\varepsilon_t = \sigma_t^* z_t$ (see Lamoureux and Lastrapes 1990 [44]; Nelson 1990 [45]) and $z_t$ is the noise process. Therefore, $\varepsilon_t^2 = \sigma_t^2 * z_t^2$.

Thus, based on this decomposition, our liquidity-adjusted GARCH model perfectly matches the decomposition of the original GARCH model. Like the GARCH-X model, the LGARCH(1,1) model uses $K_t$ to replace $\varepsilon_t^2$ but it further elaborates the $z_t$ term embedded in $\varepsilon_t$ by adopting the BAS, which is denoted as L. It can be observed from Figs 2–4 that the BAS has the similar spike clusters with both simplified realized variance and realized variance for the periods late 2007 and early 2016. Therefore, our model implies that $K_{t-1} = \varepsilon_{t-1}^2$. Similarly,

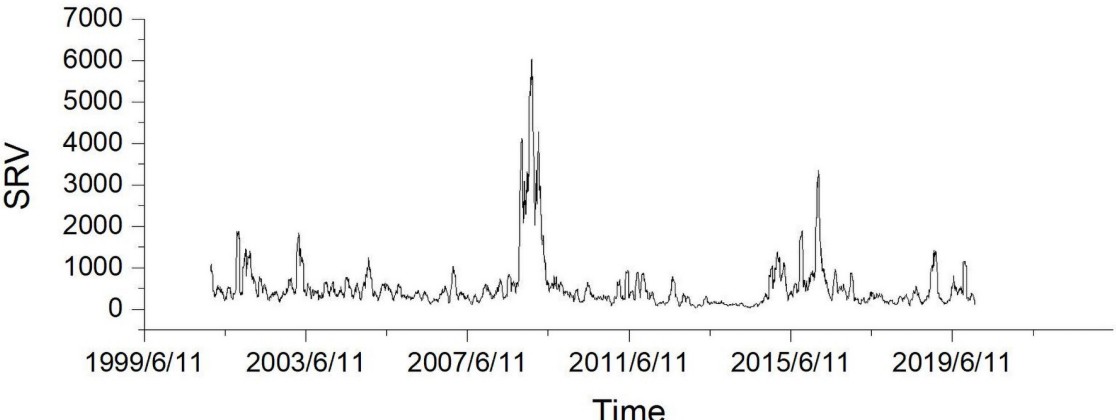

**Fig 2. Daily simplified realized variance plotting.**

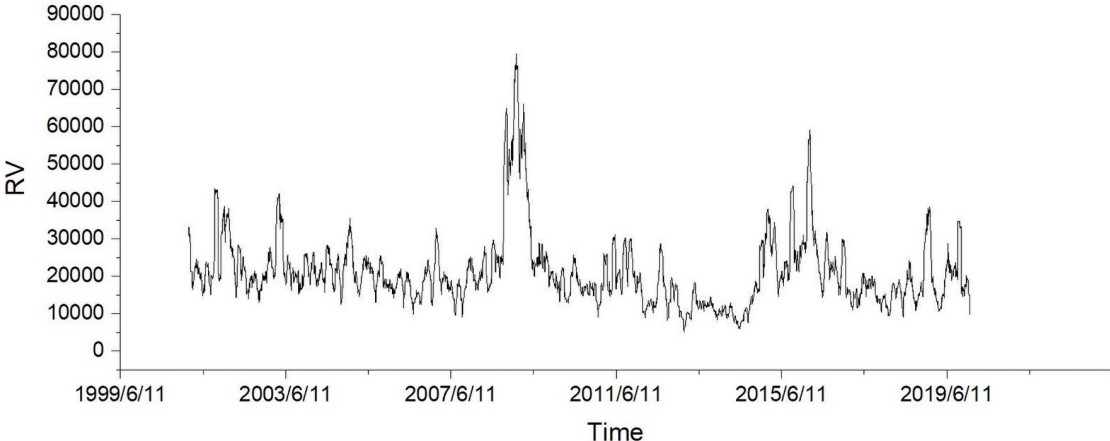

**Fig 3. Daily realized variance plotting.**

concerning the GARCH-X model, we have the general form:

$$\sigma_t^2 = \omega + \alpha y_{t-1}^2 + \beta \sigma_{t-1}^2 + \gamma x_t, \qquad (9)$$

where $x_t$ is the exogenous variable, such as interest rate and $y_t$ denotes the residual term, and $y_t^2 = \sigma_t^2 * z_t^2$, where $\sigma_t$ is the volatility of target time series and $z_t$ is the noise term, which is defined as $z_t \sim IID(0, 1)$.

So, $y_{t-1}^2 = K_{t-1}$ for the GARCH-X case. Furthermore, we have $y_t^2 = \sigma_t^2 * z_t^2$ and $z_t$ is the noise term, which is defined as $z_t \sim IID(0, 1)$. $K_t = \sigma_t^2(1 - L_t^2) * (L_t^2 - \sigma_t^2 - L_t^4)$, so, it can be further deduced as $z_t^2 = (1 - L_t^2) * (L_t^2 - \sigma_t^2 - L_t^4)$, which we denote as the liquidity adjustment (LA) on volatility. Compared with both general GARCH model as well as GARCH-X model, we have a further decomposition of the $\varepsilon_t^2$ term.

This liquidity adjustment uncovers the fundamental elements encompassed in $z_t$ and it has two states:

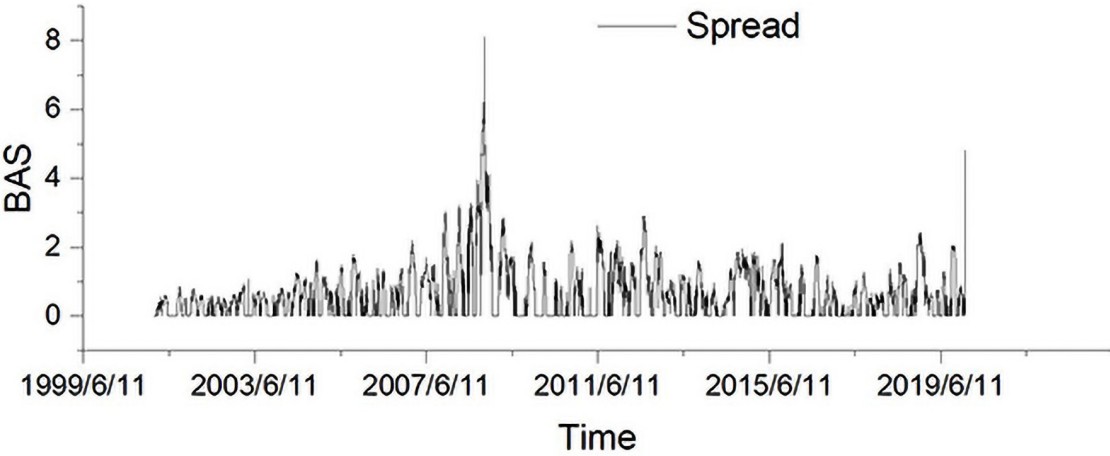

**Fig 4. Daily bid-ask spread plotting.**

1. When $L_t^2 > 1$, it indicates that the BAS in the market is relatively large and the liquidity of the market is insufficient. As a result, $(1 - L_t^2) < 0$ and $(L_t^2 - 2\sigma_t^2 - L_t^4) < 0$, which generate the positive product. Since the BAS and volatility have a positive correlation, the large BAS may indicate higher volatility in the future. Therefore, the adjustment increases the volatility term by adding a positive value to the process to reflect the potential increase of volatility in the near future.

2. When $L_t^2 < 1$, it indicates that the BAS in the market is relatively small and the market is liquid.

(i) When $L_t^2 > \sigma_t^2 + L_t^4$, the product is still positive, in which case, the spread is still not small enough. Therefore, the adjustment still has positive value to unveil the potential increase of the future volatility.

(ii) When $L_t^2 < \sigma_t^2 + L_t^4$, the product is negative, in which case, the BAS is trivial and trading frictions in the market are negligible. Therefore, the model may foresee possible volatility decrease in the near future by adding a negative value to the previous volatility.

In order to testify our model's data fitness and volatility prediction ability, we adopt two tests for the model with both in-sample and out-of-sample tests. The first test is to predict the simplified realized variance and the second test is to predict the realized variance. We are able to show that the detailed decomposition of $z_t$ in the GARCH model substantially heightens the forecasting accuracy of the GARCH model.

## 4 Empirical results

This section gives both empirical results for regression models and model performance of volatility forecasting. In particular, we compare our data fitting results as well as prediction results with three GARCH models, namely, GARCH, IGARCH and GJR-GARCH (denoted as TGARCH in the tables).

For the model performance evaluation, we use Mean Squared Error (MSE) to measure the model performance for both in-sample fitting and out-of-sample forecasting tests since our daily data can be quite noisy (Pong et al. 2004 [46]; Bollerslev et al. 2016 [47]). More importantly, MSE is quite suitable for time-series forecasting accuracy measurement (Ismail et al. 2011 [48]). The periodic averaged MSE can be defined as:

$$MSE_T = \frac{1}{T} \sum_{t=1}^{T} \left( Observed_t - Predicted_t \right)^2,$$

where T represents the number of observations embedded in the forecasting period, $Observed_t$ presents the observed variance from the market and $Predicted_t$ presents the variance predicted from the models.

Lower MSE indicates higher forecasting accuracy as well as more powerful prediction of the model.

### 4.1 Empirical models

Before the model performance evaluation, we firstly run a series of regressions to certify the significant impact of BAS on both simplified realized variance and realized variance. The

**Table 2. Regression results summary for simplified realized variance and realized variance.**

| | $SRV_t$ | $SRV_t$ | $SRV_t$ | $RV_t$ | $RV_t$ | $RV_t$ |
|---|---|---|---|---|---|---|
| $SRV_{t-1}$ | 0.175*** | 0.12*** | | 0.14*** | 0.082*** | |
| | (0.0142) | (0.033) | | (0.073) | (0.015) | |
| $L_{t-1}^2$ | 10.01*** | 6.27*** | 14.53** | 10.66*** | 11.12*** | 20.08*** |
| | (7.71) | (7.05) | (7.25) | (3.57) | (3.39) | (4.57) |
| $\varepsilon_{t-1}^2$ | | -0.01 | 0.032** | | 0.094*** | 0.002** |
| | | (0.0288) | (0.015) | | (0.013) | (0.0009) |
| $RV_{t-1}$ | | | 0.77*** | | | 1.12*** |
| | | | (0.028) | | | (0.019) |

specification of the regression model is shown in the following equations:

$$SRV_t = \beta_0 + \beta_1 SRV_{t-1} + \beta_2 \varepsilon_{t-1} + \beta_3 L_{t-1} + \beta_4 RV_{t-1} + \delta_t, \tag{10}$$

$$RV_t = \beta_0 + \beta_1 RV_{t-1} + \beta_2 \varepsilon_{t-1} + \beta_3 L_{t-1} + \beta_4 SRV_{t-1} + \delta_t. \tag{11}$$

where $SRV_t$ is the simplified realized variance, $RV_t$ is the realized variance, $L_t$ is the bid-ask spread. We run a series of multivariable regression analysis and the results are shown in Table 2. It is observable that the impact of BAS on volatility is oftentimes significant after controlling different variables. Therefore, the empirical delivers strong support of BAS's impact on volatility and we thereby incorporate it into the GARCH model for volatility forecasting.

## 4.2 In-sample data fitting

For the in-sample modeling, we compare three models with our model in fitting both simplified realized variance and realized variance. Table 3 shows the in-sample fitting MSE against the simplified realized variance for all four models during the data period 2001-2010. In general, our model outperforms other three models and the improvement rate is around 25%

**Table 3. In sample fitting with simplified realized variance.** This table presents the oil futures volatility in-sample fitting results of Liquidity-adjusted GARCH (LGARCH) with other three GARCH models against the simplified realized variance using the Mean Squared Error (MSE). The p-values for statistical differences of the forecasting errors are also presented. The LGARCH model outweighs all other three models in both full sample and most subsample tests except year 2005 compared with TGARCH. Where $en = {}^*10^n$, e.g. $e-02 = {}^*10^{-2}$.

| | | GARCH (MSE) | LGARCH (MSE) | Improve Rate (p-value) | IGARCH (MSE) | LGARCH (MSE) | Improve Rate (p-value) | TGARCH (MSE) | LGARCH (MSE) | Improve Rate (p-value) |
|---|---|---|---|---|---|---|---|---|---|---|
| Full sample | | 2.54 | 1.89 | 25.5% (0.00) | 2.56 | 1.91 | 25.3% (0.00) | 2.52 | 1.88 | 25.4% (0.00) |
| Subsamples: | Year | | | | | | | | | |
| | 2001 | 4.78 | 3.25 | 32.1% (0.00) | 4.77 | 3.25 | 32.0% (0.00) | 4.78 | 3.25 | 32.1% (0.00) |
| | 2002 | 5.23e-01 | 4.41e-01 | 15.6% (0.00) | 5.23e-01 | 4.41e-01 | 15.6% (0.00) | 5.25e-01 | 4.41e-01 | 15.9% (0.00) |
| | 2003 | 1.72 | 1.07 | 37.7% (0.00) | 1.72 | 1.07 | 37.7% (0.00) | 1.72 | 1.07 | 37.7% (0.00) |
| | 2004 | 6.2e-01 | 5.69e-01 | 8.16% (0.00) | 6.19e-01 | 5.69e-01 | 8.14% (0.00) | 6.2e-01 | 5.69e-01 | 8.16% (0.00) |
| | 2005 | 3.97e-01 | 3.91e-01 | 1.6% (0.00) | 3.98e-01 | 3.91e-01 | 1.7% (0.00) | 3.87e-01 | 3.91e-01 | -0.1% (0.00) |
| | 2006 | 2.26e-01 | 1.84e-01 | 18.8% (0.00) | 2.23e-01 | 1.84e-01 | 17.7% (0.00) | 2.26e-01 | 1.84e-01 | 18.6% (0.00) |
| | 2007 | 3.93e-01 | 3.21e-01 | 18.2% (0.00) | 3.92e-01 | 3.21e-01 | 17.9% (0.00) | 3.86e-01 | 3.21e-01 | 16.7% (0.00) |
| | 2008 | 9.24 | 8.12 | 11.9% (0.21) | 9.15 | 8.12 | 11.1% (0.30) | 9.28 | 8.12 | 12.3% (0.29) |
| | 2009 | 7.06 | 5.41 | 23.3% (0.00) | 7.09 | 5.41 | 23.6% (0.00) | 7.03 | 5.41 | 22.9% (0.00) |
| | 2010 | 2.22e-01 | 1.55e-01 | 29.8% (0.00) | 2.19e-01 | 1.55e-01 | 28.9% (0.00) | 2.16e-01 | 1.55e-01 | 27.8% (0.00) |
| Average of Subsamples | | 2.52 | 1.90 | 19.74% | 2.51 | 1.90 | 19.48% | 2.50 | 1.90 | 19.16% |

**Table 4. In sample fitting with realized variance.** This table presents the oil futures volatility in-sample fitting results of Liquidity-adjusted GARCH (LGARCH) with other three GARCH models against the realized variance using the Mean Squared Error (MSE). The p-values for statistical differences of the forecasting errors are also presented. The LGARCH model outweighs all other three models in both full sample and all subsample tests. Where $en = {}^*10^n$, e.g. $e-02 = {}^*10^{-2}$.

| | | GARCH (MSE) | LGARCH (MSE) | Improve Rate (p-value) | IGARCH (MSE) | LGARCH (MSE) | Improve Rate (p-value) | TGARCH (MSE) | LGARCH (MSE) | Improve Rate (p-value) |
|---|---|---|---|---|---|---|---|---|---|---|
| Full sample | | 5.74e-01 | 1.44e-01 | 74.9% (0.00) | 5.61e-01 | 1.44e-01 | 74.3% (0.00) | 5.65e-01 | 1.44e-01 | 74.6% (0.00) |
| Subsamples: | Year | | | | | | | | | |
| | 2001 | 3.42 | 2.47e-01 | 92.8% (0.00) | 3.44 | 2.48e-01 | 92.8% (0.00) | 3.31 | 2.48e-01 | 92.5% (0.00) |
| | 2002 | 2.63e-01 | 8.41e-02 | 96.8% (0.00) | 2.65e-01 | 8.41e-02 | 96.8% (0.00) | 2.68e-01 | 8.41e-02 | 96.9% (0.00) |
| | 2003 | 2.93 | 1.27e-01 | 95.6% (0.00) | 2.95 | 1.27e-01 | 95.6% (0.00) | 2.68 | 1.27e-01 | 95.5% (0.00) |
| | 2004 | 2.72 | 1.5e-01 | 94.4% (0.00) | 2.7 | 1.5e-01 | 94.4% (0.00) | 2.71 | 1.5e-01 | 94.5% (0.00) |
| | 2005 | 3.22 | 2.27e-01 | 92.9% (0.00) | 3.15 | 2.27e-01 | 92.8% (0.00) | 3.04 | 2.27e-01 | 92.5% (0.00) |
| | 2006 | 2.91 | 6.44e-01 | 77.8% (0.00) | 2.84 | 6.44e-01 | 77.3% (0.00) | 2.84 | 6.44e-01 | 77.5% (0.00) |
| | 2007 | 3.22e-01 | 1.11e-01 | 65.7% (0.00) | 3.16e-01 | 1.11e-01 | 65.1% (0.00) | 3.12e-01 | 1.11e-01 | 64.7% (0.00) |
| | 2008 | 2.12 | 1.89 | 10.5% (0.01) | 2.01 | 1.89 | 5.7% (0.01) | 2.07 | 1.89 | 8.4% (0.01) |
| | 2009 | 1.23 | 2.61e-01 | 23.3% (0.01) | 1.24 | 2.6e-01 | 23.6% (0.01) | 1.26 | 2.6e-01 | 22.9% (0.01) |
| | 2010 | 2.67 | 6.42e-01 | 75.8% (0.00) | 2.63 | 6.42e-01 | 75.5% (0.00) | 2.61 | 6.42e-01 | 75.2% (0.00) |
| Average of Subsamples | | 5.74e-01 | 2.48e-01 | 78.17% | 5.6e-01 | 2.48e-01 | 77.52% | 5.65e-01 | 2.48e-01 | 77.72% |

compared with other three GARCH models for the full sample test against the simplified realized variance. Moreover, the improvement rate is around 20% on average for the subsample test against the simplified realized variance. However, in the year 2005, the simplified realized variance is quite stationary (see Fig 2), our model thereby may not be able to outperform the TGARCH model since the liquidity effects on volatility might be trivial. In all other cases, our model exhibits the superior characteristics in the in-sample model fitting analysis. Additionally, Table 4 presents the in-sample fitting MSE against the realized variance for all four models during the data period 2001-2010. For the autoregressive fitting, the results of our model are dominating since our model overwhelmingly surpasses all other three models with nearly 75% improvement rate for the full sample fitting test against the realized variance. Concerning the average improvement rate of subsample tests, it is around 70% against the realized variance. This result illustrates that liquidity plays a significant role in the autoregressive model fitting for the realized variance.

## 4.3 Out-of-sample forecasting

For the out-of-sample forecasting, we also compare three models with our model in forecasting both simplified realized variance and realized variance with the prediction errors measured by MSE.

For out-of-sample forecasting, we use the normalized MSE loss function proposed by Chen and Watanabe (2019) [49]:

$$MSE_m = \frac{1}{m}\sum_{t=n+1}^{n+m}((Observed_t - Predicted_t)/2)^2, \qquad (12)$$

where m is the out-of sample size.

Table 5 shows out-of-sample forecasting MSE against the simplified realized variance for all four models during the data period 2011-2019. In general, our model outperforms other three models and the improvement rate is around 22% compared with other three GARCH models for the full sample test against the simplified realized variance. For the average improvement

**Table 5. Out of sample forecasting with simplified realized variance.** This table presents the oil futures volatility out-of-sample forecasting results of Liquidity-adjusted GARCH (LGARCH) with other three GARCH models against the simplified realized variance using the Mean Squared Error (MSE). The p-values for statistical differences of the forecasting errors are also presented. The LGARCH model outweighs all other three models in both full sample and all subsample tests. Where $en = {}^*10^n$, e.g. $e-02 = {}^*10^{-2}$.

| | | GARCH (MSE) | LGARCH (MSE) | Improve Rate (p-value) | IGARCH (MSE) | LGARCH (MSE) | Improve Rate (p-value) | TGARCH (MSE) | LGARCH (MSE) | Improve Rate (p-value) |
|---|---|---|---|---|---|---|---|---|---|---|
| Full sample | | 1.86e-01 | 1.44e-01 | 22.6% (0.00) | 1.87e-01 | 1.44e-01 | 22.9% (0.00) | 1.85e-01 | 1.44e-01 | 22.1% (0.00) |
| Subsamples: | Year | | | | | | | | | |
| | 2011 | 4.7e-01 | 4.03e-01 | 14.2% (0.00) | 4.82e-01 | 4.03e-01 | 16.3% (0.00) | 4.51e-01 | 4.03e-01 | 10.6% (0.00) |
| | 2012 | 5.17e-02 | 1.83e-02 | 64.6% (0.00) | 4.14e-02 | 1.83e-02 | 55.7% (0.00) | 5.02e-02 | 1.83e-02 | 63.5% (0.00) |
| | 2013 | 2.21e-02 | 2.05e-02 | 7.24% (0.00) | 2.18e-02 | 2.05e-02 | 5.96% (0.00) | 2.16e-02 | 2.05e-02 | 5.09% (0.00) |
| | 2014 | 3.07e-01 | 2.55e-01 | 16.9% (0.01) | 2.88e-01 | 2.55e-01 | 11.4% (0.05) | 2.74e-01 | 2.55e-01 | 6.90% (0.06) |
| | 2015 | 1.26e-01 | 9.88e-02 | 21.5% (0.00) | 1.14e-01 | 9.88e-02 | 13.3% (0.02) | 1.05e-01 | 9.88e-02 | 5.91% (0.07) |
| | 2016 | 1.84 | 1.39 | 24.4% (0.02) | 1.72 | 1.39 | 19.1% (0.03) | 1.66 | 1.39 | 16.7% (0.02) |
| | 2017 | 1.49e-01 | 1.03e-01 | 30.8% (0.00) | 1.42e-01 | 1.03e-01 | 27.4%(0.00) | 1.39e-01 | 1.03e-01 | 25.9% (0.00) |
| | 2018 | 6.21e-01 | 5.14e-01 | 17.2% (0.00) | 6.22e-01 | 5.14e-01 | 17.6%(0.00) | 5.87e-01 | 5.14e-01 | 12.5% (0.00) |
| | 2019 | 3.55e-01 | 2.54e-01 | 27.3% (0.03) | 3.34e-01 | 2.58e-01 | 22.7%(0.04) | 3.47e-01 | 2.58e-01 | 25.6% (0.05) |
| Average of Subsamples | | 4.36e-01 | 3.38e-01 | 24.95% | 4.16e-01 | 3.38e-01 | 21.11% | 4.03e-01 | 3.38e-01 | 19.22% |

rate of subsample, it is around 17% compared with the three GARCH models. Additionally, Table 6 presents the out-of-sample forecasting MSE against the realized variance for all four models during the data period 2011-2019. The results are also dominating for both full sample and subsample tests. Our model overwhelmingly surpasses all three GARCH models with the full sample improvement rate around 73% and the average improvement rate of subsample around 70%. Therefore, it is arguable that our model produces a substantially accurate forecasting results compared with all three GARCH models.

Nevertheless, for the out-of-sample forecasting, the results may not be as stable as the in sample fitting. For the data of 2013 and 2014, the oil price move trend has been changed. In 2013, the oil price had an increasing trend with the annual return of 6.90%. On the other hand, however, the oil price plummeted in 2014, with the annual return of -45.55%, which yielded

**Table 6. Out of sample forecasting with realized variance.** This table presents the oil futures volatility out-of-sample forecasting results of Liquidity-adjusted GARCH (LGARCH) with other three GARCH models against the realized variance using the Mean Squared Error (MSE). The p-values for statistical differences of the forecasting errors are also presented. The LGARCH model outweighs all other three models in both full sample and most subsample tests. Where $en = {}^*10^n$, e.g. $e-02 = {}^*10^{-2}$.

| | | GARCH (MSE) | LGARCH (MSE) | Improve Rate (p-value) | IGARCH (MSE) | LGARCH (MSE) | Improve Rate (p-value) | TGARCH (MSE) | LGARCH (MSE) | Improve Rate (p-value) |
|---|---|---|---|---|---|---|---|---|---|---|
| Full sample | | 3.41 | 9.33e-01 | 72.6% (0.00) | 3.42 | 9.33e-01 | 73.1% (0.00) | 3.56 | 9.33e-01 | 73.9% (0.00) |
| Subsamples: | Year | | | | | | | | | |
| | 2011 | 4.73 | 2.51 | 44.5% (0.03) | 4.51 | 2.51 | 44.5% (0.04) | 3.92 | 2.51 | 36.2% (0.09) |
| | 2012 | 1.55e-04 | 9.10e-05 | 49.1% (0.02) | 1.79e-04 | 9.10e-05 | 49.6% (0.02) | 1.49e-04 | 9.10e-05 | 39.1% (0.01) |
| | 2013 | 1.76e-04 | 1.43e-05 | 93.1% (0.00) | 2.05e-04 | 1.43e-05 | 93.1% (0.00) | 1.99e-04 | 1.43e-05 | 92.9% (0.00) |
| | 2014 | 2.74 | 1.57 | 42.6% (0.00) | 2.86 | 1.57 | 45.1% (0.00) | 2.71 | 1.57 | 41.8% (0.03) |
| | 2015 | 3.12e-03 | 4.42e-04 | 85.8% (0.00) | 3.05e-03 | 4.42e-04 | 85.4%(0.00) | 3.04e-03 | 4.42e-04 | 85.4% (0.00) |
| | 2016 | 2.05e-03 | 2.51e-04 | 87.7% (0.00) | 2.25e-03 | 2.51e-04 | 88.8%(0.00) | 1.55e-03 | 2.51e-04 | 83.8% (0.00) |
| | 2017 | 1.87 | 3.25e-01 | 82.5% (0.00) | 1.76 | 3.25e-01 | 81.8%(0.00) | 1.82 | 3.25e-01 | 82.4% (0.00) |
| | 2018 | 3.05e-03 | 5.24e-04 | 82.8% (0.00) | 3.20e-03 | 5.24e-04 | 83.6%(0.00) | 2.97e-03 | 5.24e-04 | 82.3% (0.00) |
| | 2019 | 4.82e-04 | 6.40e-05 | 86.7% (0.00) | 4.77e-04 | 6.40e-05 | 86.5%(0.00) | 4.54e-04 | 6.40e-05 | 85.9% (0.00) |
| Average of Subsamples | | 1.13 | 5.04e-01 | 72.0% | 1.12 | 5.04e-01 | 73.1% | 1.03 | 5.04e-01 | 69.9% |

the lowest price of $53.45 since 2010. As a result, the return volatility movement may not fol-
low similar pattern, which may result in the significant difference of volatility forecasting.

## 4.4 Robustness check

To ensure our results are robust, we also adopt the Mincer-Zarnowitz regression to verify the
relative performance of our volatility forecasting model compared with other GARCH models.
Following Mincer and Zarnowitz (1969) [50], we run following two regressions based on our
volatility forecasting results:

$$SRV_t = \beta_0 + \beta_1 \hat{v}_{model1,t} + \beta_2 \hat{v}_{model2,t} + \epsilon_t, \tag{13}$$

$$RV_t = \beta_0 + \beta_1 \hat{v}_{model1,t} + \beta_2 \hat{v}_{model2,t} + \epsilon_t, \tag{14}$$

where $SRV_t$ and $RV_t$ are the simplified realized variance and the realized variance observed at
time $t$ respectively, $\hat{v}_{model1,t}$ and $\hat{v}_{model2,t}$ are the forecasted variance from model 1 and model 2 at
time $t$ respectively.

For evaluating the model performance, we first run the single variable regression on the
value predicted by LGARCH model as the model 1 in Eqs (12) and (13). Then, we run two-vari-
able regression on both LGARCH as model 1 and one of three models, GARCH, IGARCH
and TGRACH as model 2 in Eqs (12) and (13) sequentially. Thus, we can evaluate the model
performance from two respects:

Firstly, we compare the $adj - R^2$ from single variable regression with the $adj - R^2$ from two-
variable regression to see whether the three GARCH models add significant explanatory
power to our model. Then, for the relative performance, we investigate the significance and
magnitude of the coefficients for both model 1 and model 2 in Eqs (12) and (13). Large and
significant coefficients demonstrate high performance. The model performance from Mincer-
Zarnowitz regression is shown in Tables 7–10.

In particular, Tables 7 and 8 show in-sample model performance for both simplified real-
ized variance and realized variance respectively. In Table 7, it can be seen that the $adj - R^2$ is
16.2% for our model and additional explanatory power that the three GARCH models can
enhance is quite marginal. More importantly, the coefficient of our model is much larger than
the coefficients for other three GARCH models. Similarly, from Table 8, it is observable that
the $adj - R^2$ is 58.3% for our model and the explanatory power that the three GARCH models
can add is quite small and the coefficients for other three GARCH models are all negative.

**Table 7. Regression results for simplified realized variance (in-sample).** This table reports the Mincer-Zarnowitz regression result for the model performance compari-
son regarding the in-sample simplified realized variance. SRV0 represents the single variable regression for our model only, SRV-G represents the model 1 is our model
and model 2 is GARCH model defined in Eq (12); SRV-IG represents the model 1 is our model and model 2 is IGARCH model defined in Eq (12); SRV-TG represents the
model 1 is our model and model 2 is TGARCH model defined in Eq (12). *, **, *** indicate statistical significance at 10%, 5% and 1% levels, respectively.

| | SRV0 | SRV-G | SRV-IG | SRV-TG |
|---|---|---|---|---|
| LGARCH | 1.03*** | 1.17*** | 1.16*** | 1.19*** |
| | (0.046) | (0.057) | (0.057) | (0.058) |
| GARCH | | -0.31*** | | |
| | | (0.077) | | |
| IGARCH | | | -0.30*** | |
| | | | (0.079) | |
| TGARCH | | | | -0.334*** |
| | | | | (0.078) |
| $Adj - R^2$ | 0.163 | 0.168 | 0.167 | 0.169 |

**Table 8. Regression results for realized variance (in-sample).** This table reports the Mincer-Zarnowitz regression result for the model performance comparison regarding the in-sample realized variance. RV0 represents the single variable regression for our model only, RV-G represents the model 1 is our model and model 2 is GARCH model defined in Eq (13); RV-IG represents the model 1 is our model and model 2 is IGARCH model defined in Eq (13); RV-TG represents the model 1 is our model and model 2 is TGARCH model defined in Eq (13). *, **, *** indicate statistical significance at 10%, 5% and 1% levels, respectively.

|  | RV0 | RV-G | RV-IG | RV-TG |
|---|---|---|---|---|
| LGARCH | 0.0079*** | 0.0073*** | 0.0073*** | 0.0073*** |
|  | (0.00013) | (0.00016) | (0.00016) | (0.00017) |
| GARCH |  | 0.0013*** |  |  |
|  |  | (0.00022) |  |  |
| IGARCH |  |  | 0.0014*** |  |
|  |  |  | (0.00021) |  |
| TGARCH |  |  |  | 0.0013*** |
|  |  |  |  | (0.00023) |
| $Adj - R^2$ | 0.583 | 0.589 | 0.588 | 0.586 |

Therefore, our model overwhelmingly outperforms the other three GARCH models regarding the in-sample volatility forecasting.

On the other hand, Tables 9 and 10 exhibit out-of-sample model performance for both simplified realized variance and the realized variance. Table 9 presents the $adj - R^2$ is 9.47% for forecasting the simplified realized variance regarding the single regression of our model. The additional $adj - R^2$ the other three GARCH models can add is nearly 2% and the coefficients are much smaller than our model. Similarly, Table 10 represents the $adj - R^2$ is 62.8% for forecasting the realized variance regarding the single regression of our model. The additional $adj - R^2$ the other three GARCH models can add is nearly 4%. The coefficient for our model is quite close to the IGARCH model, but is still larger than other two GARCH models. As a result, our model in general outperforms the other three GARCH models for the out-of-sample volatility forecasting.

Finally, we employ the QLIKE loss function for the robustness purpose. The QLIKE loss function could be written as:

$$QLIKE_j = \frac{1}{T}\sum_{t=1}^{T}\left(ln\hat{\sigma}_{t,j}^2 + \frac{\sigma_{t,j}^2}{\hat{\sigma}_{t,j}^2}\right), \tag{15}$$

where $\sigma_{t,j}^2$ is the real value of variance and $\hat{\sigma}_{t,j}^2$ is the predicted value of variance. From Table 11,

**Table 9. Regression results for simplified realized variance (out-of-sample).** This table reports the Mincer-Zarnowitz regression result for the model performance comparison regarding the out-of-sample simplified realized variance. SRV0 represents the single variable regression for our model only, SRV-G represents the model 1 is our model and model 2 is GARCH model defined in Eq (12); SRV-IG represents the model 1 is our model and model 2 is IGARCH model defined in Eq (12); SRV-TG represents the model 1 is our model and model 2 is TGARCH model defined in Eq (12). *, **, *** indicate statistical significance at 10%, 5% and 1% levels, respectively.

|  | SRV0 | SRV-G | SRV-IG | SRV-TG |
|---|---|---|---|---|
| LGARCH | 0.68*** | 0.53*** | 0.54*** | 0.55*** |
|  | (0.053) | (0.058) | (0.057) | (0.058) |
| GARCH |  | 0.061*** |  |  |
|  |  | (0.098) |  |  |
| IGARCH |  |  | 0.059*** |  |
|  |  |  | (0.095) |  |
| TGARCH |  |  |  | 0.057*** |
|  |  |  |  | (0.096) |
| $Adj - R^2$ | 0.0947 | 0.115 | 0.116 | 0.114 |

**Table 10. Regression results for realized variance (out-of-sample).** This table reports the Mincer-Zarnowitz regression result for the model performance comparison regarding the out-of-sample realized variance. RV0 represents the single variable regression for our model only, RV-G represents the model 1 is our model and model 2 is GARCH model defined in Eq (13); RV-IG represents the model 1 is our model and model 2 is IGARCH model defined in Eq (13); RV-TG represents the model 1 is our model and model 2 is TGARCH model defined in Eq (13). *, **, *** indicate statistical significance at 10%, 5% and 1% levels, respectively.

|  | RV0 | RV-G | RV-IG | RV-TG |
|---|---|---|---|---|
| LGARCH | 0.005*** | 0.0039*** | 0.002*** | 0.005*** |
|  | (0.00025) | (0.0003) | (0.0006) | (0.00003) |
| GARCH |  | 0.001*** |  |  |
|  |  | (0.0003) |  |  |
| IGARCH |  |  | 0.002*** |  |
|  |  |  | (0.0005) |  |
| TGARCH |  |  |  | 0.0003* |
|  |  |  |  | (0.00019) |
| $Adj - R^2$ | 0.628 | 0.666 | 0.667 | 0.63 |

we found that the QLIKE loss function results for our GARCH model is always the lowest among the four models. The QLIKE loss function results are robust for in-sample periods.

For out-of-sample forecasting, we use the normalized QLIKE loss function proposed by Chen and Watanabe (2019)

$$QLIKE_j = \frac{1}{m} \sum_{t=n+1}^{n+m} \left( \frac{\sigma_{t,j}^2}{\hat{\sigma}_{t,j}^2} - ln \left( \frac{\sigma_{t,j}^2}{\hat{\sigma}_{t,j}^2} \right) - 1 \right), \tag{16}$$

where $m$ is the out-of sample size, $\hat{\sigma}_{t,j}^2$ is the predicted value and $\sigma_{t,j}^2$ is the actual value.

We compare our model with both ANN-GARCH and SVM-GARCH models, reporting the results in Table 12. The ANN-GARCH model is based on the Artificial Neural Networks, which is the non-parametric method. This method can be applied to uncover the nonlinear associations between the parameters of the GARCH model. Nevertheless, ANN-GARCH is less flexible compared with other models such as neural fuzzy inference system. In addition, the backpropagation algorithm of the ANN disables itself to learn from its own forecasting error. (Kristjanpoller and Minutolo, 2016 [51]).

For the ANN-GARCH method, each neural network connects a group of volatility forecasting variables with output variables and a number of hidden layers. Neurons are connected between the layers for connections that are activated by triggering a threshold. The input group of variables and output group of variables can be a combination of all of different number of neurons in each layer, where the relations between inputs and outputs are embedded (see Kristjanpoller and Hernandez, 2017 [52]; Bhattacharya and Ahmed, 2018 [53]).

The SVM-GARCH model is based on the support vector machine (SVM) stems from the statistical learning theory (see Cortes and Vapnik, 1995 [54]), For the SVM method, it develops

**Table 11. QLIKE loss function results for four models.** This table reports the QLIKE loss function results for the model performance comparison regarding the four models. SRV represents the simplified realized variance, IS represents the in-sample results (from year 2000-2010) and OS represents the out-of-sample results (from year 2011-2019).

|  | RV-IS | SRV-IS | RV-OS | SRV-OS |
|---|---|---|---|---|
| LGARCH | -12.96 | -7.24 | 10.87 | 3.11 |
| GARCH | -10.73 | -6.54 | 19.24 | 3.86 |
| IGARCH | -10.65 | -6.55 | 19.38 | 3.89 |
| TGARCH | -10.59 | -6.56 | 18.88 | 3.88 |

**Table 12. QLIKE loss function results for ANN and SVM models.** This table reports the QLIKE loss function results for the model performance comparison regarding our model with GARCH models with ANN and SVM. SRV represents the simplified realized variance, IS represents the in-sample results (from year 2000-2010) and OS represents the out-of-sample results (from year 2011-2019). SVMl, SVMp and SVMg represent the SVM with linear, polynomial and Gaussian kernel, respectively.

| | RV-IS | SRV-IS | RV-OS | SRV-OS |
|---|---|---|---|---|
| LGARCH | -12.96 | -7.24 | 10.87 | 3.11 |
| ANN-GARCH | -11.04 | -7.02 | 18.11 | 3.55 |
| SVMl-GARCH | -10.53 | -6.22 | 19.07 | 3.78 |
| SVMp-GARCH | -10.85 | -6.67 | 18.89 | 3.69 |
| SVMg-GARCH | -10.96 | -6.84 | 18.75 | 3.62 |

a nonlinear mapping function from input space towards a high-dimensional hidden space. On the basis, SVM estimates a linear regression model in the output space, which corresponds to a nonlinear regression in the low-dimensional input space. Theoretically, SVM could approximate any nonlinear mapping relations between input space and output space.

Nevertheless, since SVM has different key kernels, for instance, linear kernel, polynomial kernel, and Gaussian kernel, the choice of kernels may become tricky. For example, if the residuals of regression model are Gaussian, the results that SVM delivers may not be solid enough, since this kernel is based on Probability Distribution Function, residuals generated by SVM may not follow Gaussian distribution. (see Perez-Cruz et al., 2003 [55]; Chen et al., 2010 [56]).

Based on Table 12, it is thereby arguable that our GARCH outperforms other three GARCH models and we are confident that our results are robust. Further in Table 12, we adopt the QLIKE function to compare our model with the GARCH models with Artificial Neural Network (ANN) and Support Vector Machine (SVM). For SVM-GARCH, we have employed different kernel functions, including linear, polynomial and Gaussian kernels. By comparing our model with those models, we also demonstrate that our results are robust since our model has the lowest QLIKE values.

## 5 Conclusion and implications

To sum up, fin-tech is a popular financial topic more recently and with the aid from computer science, we are able to improve GARCH family models, which are widely used model in financial time series analysis. This paper proposes the improved GARCH model, which integrates the liquidity factor. Our new GARCH model, named LGARCH, takes the bid-ask spread term into the volatility forecasting, shedding insights on the understanding of the GARCH model as well as the impact of liquidity on the volatility forecasting process. Since how to integrate liquidity into the volatility forecasting model is open to discussion, we use GP method to identify the specific model format for the LGARCH model. This result also indicates that the liquidity factor plays a vital role in oil volatility forecasting since it appears in the final model format. In this vein, fintech facilitates our understanding of oil volatility forecasting as a new factor has been introduced.

Compared with GARCH, IGARCH and TGARCH, our model generally outperforms all three models for predicting both simplified realized variance and realized variance. More importantly, the improvement rate of our model compared with three GARCH models is around 20% for forecasting simplified realized variance and 70% for forecasting realized variance. We also use the Mincer-Zarnowitz regression to demonstrate our results are robust. The strong empirical results and outstanding model performance deliver significant findings on liquidity impact on volatility and yields helpful techniques for volatility forecasting. In

addition, the more accurate forecasting model can be integrated into the existing fintech systems such as high frequency trading platforms and derivative trading platforms by providing better volatility estimation. Our sample period has 16 years oil daily data, which covered before and after financial crisis periods. Moreover, our paper has two robustness measures for both in-sample and out-of-sample periods, demonstrating the generality and validity of our results.

Furthermore, our LGARCH model can also yield a couple of policy implications. Firstly, compared with traditional GARCH models, our oil volatility forecasting model can provide more reliable volatility of crude oil market. The reliable volatility prediction can help oil importing economies to determine countries' oil reserve levels in order to alleviate the negative impact on the economy. More importantly, market volatility might reflect the fragility of financial markets and economy (Celik and Ergin, 2014 [57]). Therefore, reliable forecasts of oil prices' volatility may play crucial roles for macroeconomic policy makers in setting monetary policies to stabilize the economy. Since our results have shown better volatility prediction of the crude oil market, energy economists, energy policy makers, and financial analysts may include market liquidity effect into the volatility forecasting models and utilize market liquidity as an indicator for future volatility changes.

Finally, Genetic Programming (GP) is the subdiscipline of evolutionary algorithms and it optimizes forecasting functions according to an evolutionary process with the evaluation criterion nested. We need to emphasize that, this paper is just a demonstration of using GP into volatility forecasting for the oil market, which can serve as a starting point of fintech development. The GP system in volatility forecasting can be further applied into other financial markets, such as stock market and foreign exchange market. More importantly, since GP is good at providing novel and unexpected insights, it can also be applied to other financial forecasting problems, such as return forecasting, price forecasting and correlation forecasting. As a result, it could be just a starting point of the technology integration in financial modelling for fintech.

## Author Contributions

**Conceptualization:** Shusheng Ding, Tianxiang Cui.

**Data curation:** Shusheng Ding.

**Formal analysis:** Tianxiang Cui.

**Methodology:** Tianxiang Cui.

**Project administration:** Jiawei Li.

**Visualization:** Yongmin Zhang.

**Writing – original draft:** Shusheng Ding.

**Writing – review & editing:** Yongmin Zhang, Jiawei Li.

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
