## [Decision Letter · Decision Letter 0]

12 Jul 2021

PONE-D-21-17939

Liquidity Effects on Oil Volatility Forecasting: From Fintech Perspective

PLOS ONE

Dear Dr. Cui,

Thank you for submitting your manuscript to PLOS ONE. After careful consideration, we feel that it has merit but does not fully meet PLOS ONE’s publication criteria as it currently stands. Therefore, we invite you to submit a revised version of the manuscript that addresses the points raised during the review process.

The authors use the period from January 1, 2001 to December 31, 2010 as the in-sample period and the period from January 1, 2011 to December 31, 2016 as the out-of-sample period. However, the data is too old. The authors are recommended to extend the out-of-sample period to 2019 at least and to update the results.

The authors should normalize both loss functions, MSE and QLIKE, to be the robust and homogeneous loss functions proposed by Patton (2011); see Chen and Watanabe (2019) for details.

We look forward to receiving your revised manuscript.

Kind regards,

Cathy W.S. Chen, Ph.D.

Academic Editor

PLOS ONE

Additional Editor Comments (if provided):

The authors use the period from January 1, 2001 to December 31, 2010 as the in-sample period and the period from January 1, 2011 to December 31, 2016 as the out-of-sample period. However, the data is too old. The authors are recommended to extend the out-of-sample period to 2019 at least and to update the results.

The authors should normalize both loss functions, MSE and QLIKE, to be the robust and homogeneous loss functions proposed by Patton (2011); see Chen and Watanabe (2019) for details.

Chen, C.W.S. and Watanabe, T. (2019) Bayesian modeling and forecasting of Value-at-Risk via threshold realized volatility, Applied Stochastic Models in Business and Industry, 35, 747-765.

Patton A.J. (2011) Volatility forecast comparison using imperfect volatility proxies. Journal of Econometrics, 160, 246-256.

Journal Requirements:

Reviewers' comments:

Reviewer's Responses to Questions

**Comments to the Author**

1. Is the manuscript technically sound, and do the data support the conclusions?

Reviewer #1: Partly

Reviewer #2: Yes

2. Has the statistical analysis been performed appropriately and rigorously? 

Reviewer #1: No

Reviewer #2: Yes

3. Have the authors made all data underlying the findings in their manuscript fully available?

Reviewer #1: Yes

Reviewer #2: Yes

4. Is the manuscript presented in an intelligible fashion and written in standard English?

Reviewer #1: Yes

Reviewer #2: Yes

5. Review Comments to the Author

Reviewer #1: This paper proposed a liquidity-adjusted GARCH model, denoted by LGARCH, for volatility forecasting by employing the Genetic Programming (GP) technique. An empirical study is conducted to investigate the fitting and prediction performances of the LGARCH model compared with the GARCH, IGARCH, and TGARCH models. The authors claimed that the LGARCH model overwhelmingly surpasses all the three GARCH models in both in-sample volatility fitting and out-of-sample volatility forecasting.

I agree that incorporating AI techniques is an exciting direction for modeling financial time series. This study belongs to this direction and focuses on including liquidity effects on oil volatility forecasting. However, I have several concerns and do not recommend the manuscript for publication in its current state. Please see the details of my report in the attachment.

Reviewer #2: In order to forecast the oil volatility with liquidity, the authors propose the Genetic programming (GP) method to identify the model format. Based on the GP method, the authors propose the liquidity-adjusted GARCH (LGARCH) model. For comparison, they consider the GARCH, IGARCH, and GJR-GARCH models. The empirical results show that the proposed LGARCH model outperforms in both sample fitting and out of sample forecasting. The robust checking is guaranteed using the loss function QLIKE. The results are interesting but the authors need to revise the paper carefully such as undefined random errors or some functions. The comments and questions are as follows. In the hope that they will be helpful to the authors.

See the attached file for the comments.

6. PLOS authors have the option to publish the peer review history of their article (what does this mean?). If published, this will include your full peer review and any attached files.

Reviewer #1: No

Reviewer #2: No

---

## [Author Response · Author response to Decision Letter 0]

26 Jul 2021

(Also see attached file)

Comments from Reviewer 1

1. In the GP method, the authors should explain the method in details. What is the criterion for choosing the best model? For example, in line 226, the definition of r2 should be clarified. The selection basis should be clarified. For instance, in equation (6), is there any regular condition for the function f? Is there any penalty function to avoid the overfitting problem? 

Response: Thanks, the selection criterion has been further clarified. Normally, the ending conditions for evolutionary computation based method are a desired solution is found (the measure of function fitness approximates 0), or the maximum number of generations is reached. For this work, as the fitness measure will never reach 0, therefore the system will terminate after a certain pre-defined number of generations (100 in this work). The parameters settings used here are based on empirical experiences and we do not claim that these settings are the optimal choices. Our main purpose is to test the effectiveness of GP approach on volatility forecasting task. r2 can be defined as the squared return term, which is used to approximate the volatility. We do not use the regular condition for function f, in order to prevent overfitting, we further eliminate the forecasting functions with more than 10 terms obtained by GP, which is similar to the idea of dropout in neural network training. 

2. In empirical analysis, the authors could explain the results in details. For example, in Table 5, the MSE for the out of sample forecasting with simplified realized variance improve around 58% in 2012 but improve only 2% in 2013. On the other hand, the MSE for the out of sample forecasting with realized variance improve improves 98% in 2013 but only improves around 40% in 2014. Is there any significant difference between the above data? 

Response: Nevertheless, for the out-of-sample forecasting, the results may not be as stable as the in sample fitting. For the data of 2013 and 2014, the oil price move trend has been changed. In 2013, the oil price had an increasing trend with the annual return of 6.90%. On the other hand, however, the oil price plummeted in 2014, with the annual return of -45.55%, which yielded the lowest price of $53.45 since 2010. As a result, the return volatility movement may not follow similar pattern, which may result in the significant difference of volatility forecasting. 

3. In addition, in empirical studies, do the authors consider the rolling method for the estimation? The authors should clarify this. 

Response: we have considered the rolling over method. For the out-of-sample forecasting, we use previous year’s data to forecast the volatility of this year. For example, we use the data of 2013 to estimate the model parameters and then use those parameters to forecast volatility of year 2014. Similarly, we use the data of 2014 to estimate the model parameters and then use those parameters to forecast volatility of year 2015. We thereby roll over the sample by one year.

4. In equation (3), (10), and (11), the definition of η t is missing. The authors should clarify this. 

Response: ηt has been defined.

5. On page 5, the sentence from line 171 to line 173 “He further assumes that Qt is equally likely to be 171 +1 first difference of equation and plugs in the result from equation” is not clear. The authors should revise this. 

Response: Thank you for your comments, we have corrected the mistake.

6. On page 5, is COV identical to cov? The notations have to be consistent. 

Response: Thank you for your comments, we have corrected the mistake.

7. On page 7, in line 245, z t is not defined clearly. 

Response: zt has been defined

8. In equation (14), the authors propose the loss function QLIKE. However, the definition is not clear. The robustness implies that the term ˆσ t,j σ t,j is close to one but there is no constraint for the first term ln ˆσ t,j . Hence, the authors should clarify why the smaller QLIKE, the better performance. 

Response: To further justify the issue with QLIKE, we adopted the normalize loss function for QLIKE (see equation (16)), which is mainly used for out-of-sample test.

9. Some notations must be Italic such as in line 170 “E” and in line 350 and line 351 “t”. 

Response: Thank you for your comments, we have corrected the mistake.

Comments from Reviewer 2

1. Since volatility cannot be observed directly, this study employs the SRVt in (1) (or RV t in (2)) as a proxy of volatility. The objective of the GP introduced in Section 3.2 is designed to minimize the error between the SRVt (or RVt) and the associated predicted variance. This design is highly related to the evaluation measure MSE T defined in Section 4, where the Observed t is the SRVt (or RVt) and the Predicted t is the predicted variance. However, for the three considered GARCH models, the model parameters are estimated under different criteria. For example, they aim to minimize the L 2 loss between the observed returns and the associated predicted returns. In other words, the three GARCH models are not designed to produce volatilities close to SRV t (or RVt ) but to capture the heteroscedasticity in the return process. The role of the volatility obtained from the three GARCH models is the conditional variance of r t 1 conditional on the information up to t−1. Therefore, it is not surprising that their volatilities are not close to SRV t (or RVt ). 

Due to these facts, I would say that the comparison study in this article is not fair, and the conclusion that “the LGARCH model overwhelmingly surpasses all the three GARCH models” is not proper since the LGARCH and the three GARCH models are designed for different targets. Most importantly, the current comparison is difficult to identify whether the good performance of the LGARCH model is due to the liquidity effects or just because of the different estimation targets of the LGARCH and the three GARCH models. 

Response: There are two reasons why the comparison is fair. Firstly, although the GARCH model is designed to capture the heteroscedasticity in the return process, the purpose of capturing the heteroscedasticity is to accurately forecast the volatility. The heteroscedasticity means the volatility of the random variable varies with time. The GARCH model is designed to capture such volatility movement pattern in order to predict its future path. More importantly, our model is based on GARCH model and the format is quite similar with the original GARCH model, which cannot have huge estimation target difference compared with the original GARCH model.

Furthermore, for the purpose of capturing the dynamics of volatility, the family of GARCH models has been widely applied to energy markets such as oil market. Sadorsky (2006) for instance, finds that the threshold GARCH (or GJR, Glosten et al., 1993) fits well for heating oil and natural gas volatilities, whereas the standard GARCH(1,1) model fits well for crude oil and unleaded gasoline volatilities. Kang et al. (2009) show that component GARCH (CGARCH) and fractional integrated GARCH (FIGARCH) provide superior performance. Recently, Ma et al. (2021) prove GARCH family models fit oil volatility quite smoothly. Liang et al. (2021) also demonstrate that GARCH-MIDAS-type models provide solid performance for natural gas futures volatility forecasting. 

2. In the literature, many other approaches are designed for estimating proxies of volatility similar to SRV t (or RV t ), but without considering liquidity effects. For example, Chen et al. (2010) proposed a GARCH-SVM approach for volatility fore- casting and adopted a proxy of volatility similar to RV t . Hajizadeh et al. (2012) proposed an ANN-GARCH type model for forecasting the volatility of S&P 500 index return and employed a proxy of volatility similar to RV t . I suggest the authors design a new comparison scenario to investigate whether the LGARCH outperforms these types of approaches and whether including the liquidity effects improves the volatility forecasting. 

Response: Thank you for your comments, we have now included the ANN-GARCH and GARCH-SVM model comparison in our robustness check part.

3. Many notations and relationships between random variables are not well defined. Some examples are listed in the following: 

(a) Line 151: Pt is not defined. 

Response: Pt has been defined. 

(b) Line 151 and 169: Do Pt and St represent the same thing? 

Response: Not exactly, Pt is the last day's settlement price and St is last observed traded price.

(c) Line 167: ηt is not defined. 

Response: ηt has been defined. 

(d) Line 167 and 169: The relationship between η t and Q t is not clear. Are they independent or uncorrelated? 

Response: the relationship has been defined. 

(e) Line 176: How to estimate the ‘spread’ from data is not clear. 

Response: it has been clarified.

(f) Line 178: The ‘residual term’ mentioned here is not clearly defined. 

Response: The ‘residual term’ has been clarified.

(g) Line 245: zt is not defined. 

Response: zt has been defined

(h) Line 254 and 255: yt−1 is not defined. If yt denotes the day-t return, I suggest to replace it by rt as in (1) and (2) for consistency purposes. 

Response: Not exactly, yt denotes the residual term, which contains the noise information that has not been captured by the GARCH-X model. yt-1 has been defined now.

According to the above reasons, I do not recommend the manuscript for publication in its current state 

Additional Comments from Editor

The authors use the period from January 1, 2001 to December 31, 2010 as the in-sample period and the period from January 1, 2011 to December 31, 2016 as the out-of-sample period. However, the data is too old. The authors are recommended to extend the out-of-sample period to 2019 at least and to update the results.

Response: Thank you for your comments, we have updated our results to the year of 2019.

The authors should normalize both loss functions, MSE and QLIKE, to be the robust and homogeneous loss functions proposed by Patton (2011); see Chen and Watanabe (2019) for details.

Response: Thank you for your comments, we have used the normalize loss functions, for both MSE and QLIKE.

Chen, C.W.S. and Watanabe, T. (2019) Bayesian modeling and forecasting of Value-at-Risk via threshold realized volatility, Applied Stochastic Models in Business and Industry, 35, 747-765.

Patton A.J. (2011) Volatility forecast comparison using imperfect volatility proxies. Journal of Econometrics, 160, 246-256.

---

## [Decision Letter · Decision Letter 1]

16 Sep 2021

PONE-D-21-17939R1

Liquidity Effects on Oil Volatility Forecasting: From Fintech Perspective

PLOS ONE

Dear Dr. Cui,

Thank you for submitting your manuscript to PLOS ONE. After careful consideration, we feel that it has merit but does not fully meet PLOS ONE’s publication criteria as it currently stands. Therefore, we invite you to submit a revised version of the manuscript that addresses the points raised during the review process.

1)      All values shown in Tables 3 - 6 are extremely small.  This situation can be improved by rescaling the time series {r_t}. Researchers use the following formula for the return.

Line 156: R_t=(ln(p_t)-ln(p_{t-1}) )\\times 100.

2)      Eq. (2) should be rescaled too. Most related papers adopt the following formula for realized volatility in order to avoid small values.

RV*_t= (RV_t \\times 10,000) or

RV**_t=ln(RV_t \\times 10,000).

We look forward to receiving your revised manuscript.

Kind regards,

Cathy W.S. Chen, Ph.D.

Academic Editor

PLOS ONE

Journal Requirements:

Additional Editor Comments (if provided):

Thank you for responding to my previous comments. This revision needs to be improved in the following ways before it can be published.

1) All values shown in Tables 3 - 6 are extremely small. This situation can be improved by rescaling the time series {r_t}. Researchers use the following formula for the return.

Line 156: R_t=(ln(p_t)-ln(p_{t-1}) )\\times 100.

2) Eq. (2) should be rescaled too. Most related papers adopt the following formula for realized volatility in order to avoid small values.

RV*_t= (RV_t \\times 10,000) or

RV**_t=ln(RV_t \\times 10,000).

Reviewers' comments:

Reviewer's Responses to Questions

**Comments to the Author**

1. If the authors have adequately addressed your comments raised in a previous round of review and you feel that this manuscript is now acceptable for publication, you may indicate that here to bypass the “Comments to the Author” section, enter your conflict of interest statement in the “Confidential to Editor” section, and submit your "Accept" recommendation.

Reviewer #1: (No Response)

Reviewer #2: All comments have been addressed

2. Is the manuscript technically sound, and do the data support the conclusions?

Reviewer #1: Partly

Reviewer #2: Yes

3. Has the statistical analysis been performed appropriately and rigorously? 

Reviewer #1: No

Reviewer #2: Yes

4. Have the authors made all data underlying the findings in their manuscript fully available?

Reviewer #1: Yes

Reviewer #2: Yes

5. Is the manuscript presented in an intelligible fashion and written in standard English?

Reviewer #1: Yes

Reviewer #2: Yes

6. Review Comments to the Author

Reviewer #1: This revision does not satisfactorily address some of my previous questions. Please see my report in the attachment.

Reviewer #2: The paper is now well written. However, there are still several comments as follows. See the attached file.

7. PLOS authors have the option to publish the peer review history of their article (what does this mean?). If published, this will include your full peer review and any attached files.

Reviewer #1: No

Reviewer #2: No

---

## [Decision Letter · Decision Letter 2]

18 Oct 2021

PONE-D-21-17939R2Liquidity Effects on Oil Volatility Forecasting: From Fintech PerspectivePLOS ONE

Dear Dr. Cui,

Thank you for submitting your manuscript to PLOS ONE. After careful consideration, we feel that it has merit but does not fully meet PLOS ONE’s publication criteria as it currently stands. Therefore, we invite you to submit a revised version of the manuscript that addresses the points raised during the review process.

I provided two comments for the previous version. However, the authors do not follow my suggestions for the returns.  We can see the range of returns is (-0.1654, 0.1640) in Table 1, which does not follow the formula. Please see the attached file. The authors will have one last chance to revise it and all related results accordingly.

We look forward to receiving your revised manuscript.

Kind regards,

Cathy W. S. Chen, Ph.D.

Academic Editor

PLOS ONE

Journal Requirements:

Additional Editor Comments (if provided):

AE

To be clear and honest, I am not satisfied with the authors’ response. I provided two comments for the previous version. However, the authors do not follow my suggestions for the returns. We can see the range of returns is (-0.1654, 0.1640) in Table 1, which does not follow the formula. Please see the attached file. The authors will have one last chance to revise it and all related results accordingly.

Reviewers' comments:

Reviewer's Responses to Questions

**Comments to the Author**

1. If the authors have adequately addressed your comments raised in a previous round of review and you feel that this manuscript is now acceptable for publication, you may indicate that here to bypass the “Comments to the Author” section, enter your conflict of interest statement in the “Confidential to Editor” section, and submit your "Accept" recommendation.

Reviewer #1: All comments have been addressed

Reviewer #2: (No Response)

Reviewer #3: (No Response)

2. Is the manuscript technically sound, and do the data support the conclusions?

Reviewer #1: Yes

Reviewer #2: Yes

Reviewer #3: Yes

3. Has the statistical analysis been performed appropriately and rigorously? 

Reviewer #1: Yes

Reviewer #2: Yes

Reviewer #3: Yes

4. Have the authors made all data underlying the findings in their manuscript fully available?

Reviewer #1: Yes

Reviewer #2: Yes

Reviewer #3: Yes

5. Is the manuscript presented in an intelligible fashion and written in standard English?

Reviewer #1: Yes

Reviewer #2: Yes

Reviewer #3: Yes

6. Review Comments to the Author

Reviewer #1: The authors have satisfactorily responded to all my questions in this revision. Nevertheless, I find one typo and have a minor comment for the revision. Please see the details in the attachment. I recommend publication of this work after the authors correct the typo.

Reviewer #2: The paper is well written now. However, there is one more issue for the paper. The authors should clarify this. See the attached file.

Reviewer #3: I have no further comment on this revised manuscript. I think this paper is good enough to be published.

7. PLOS authors have the option to publish the peer review history of their article (what does this mean?). If published, this will include your full peer review and any attached files.

Reviewer #1: No

Reviewer #2: No

Reviewer #3: No

---

## [Author Response · Author response to Decision Letter 2]

25 Oct 2021

Thank you for your comments and we have now rescaled the returns and volatility estimation based on the suggestions. Sorry for the mistake in the

last manuscript.

---

## [Decision Letter · Decision Letter 3]

8 Nov 2021

Liquidity Effects on Oil Volatility Forecasting: From Fintech Perspective

PONE-D-21-17939R3

Dear Dr. Cui,

We’re pleased to inform you that your manuscript has been judged scientifically suitable for publication and will be formally accepted for publication once it meets all outstanding technical requirements.

Kind regards,

Cathy W. S. Chen, Ph.D.

Academic Editor

PLOS ONE

Additional Editor Comments (optional):

Reviewers' comments:

Reviewer's Responses to Questions

**Comments to the Author**

1. If the authors have adequately addressed your comments raised in a previous round of review and you feel that this manuscript is now acceptable for publication, you may indicate that here to bypass the “Comments to the Author” section, enter your conflict of interest statement in the “Confidential to Editor” section, and submit your "Accept" recommendation.

Reviewer #2: All comments have been addressed

2. Is the manuscript technically sound, and do the data support the conclusions?

Reviewer #2: Yes

3. Has the statistical analysis been performed appropriately and rigorously? 

Reviewer #2: Yes

4. Have the authors made all data underlying the findings in their manuscript fully available?

Reviewer #2: Yes

5. Is the manuscript presented in an intelligible fashion and written in standard English?

Reviewer #2: Yes

6. Review Comments to the Author

Reviewer #2: The paper is now well written. The authors have fully addressed the points raised in the referee report.

7. PLOS authors have the option to publish the peer review history of their article (what does this mean?). If published, this will include your full peer review and any attached files.

Reviewer #2: No

---

## [Editor Report · Acceptance letter]

17 Nov 2021

PONE-D-21-17939R3 

Liquidity Effects on Oil Volatility Forecasting: From Fintech Perspective 

Dear Dr. Cui:

I'm pleased to inform you that your manuscript has been deemed suitable for publication in PLOS ONE. Congratulations! Your manuscript is now with our production department. 

Kind regards, 

on behalf of

Prof. Cathy W. S. Chen 

Academic Editor

PLOS ONE